# COVID-19 and self-reported health of the Norwegian adult general population: A longitudinal study 3 months before and 9 months into the pandemic

Andrew M. Garratt[1]*, Knut Stavem[2,3,4]

1 Division for Health Services, Norwegian Institute of Public Health, Oslo, Norway, 2 Health Services Research Unit, Akershus University Hospital, Lørenskog, Norway, 3 Institute of Clinical Medicine, University of Oslo, Oslo, Norway, 4 Medical Division, Department of Pulmonary Medicine, Akershus University Hospital, Nordbyhagen, Norway

* andrew.garratt@fhi.no

**Data Availability Statement:** Data are publicly available at OSF (https://osf.io/8zwem/files/osfstorage) under the file name NorwayGenPopC19.csv.

## Abstract

The COVID-19 pandemic had a global impact on daily lives, and this study aimed to assess the effects on broader aspects of health in the general population of Norway. This population-based cohort study assessed changes in health of the Norwegian general population from 3 months before to 9 months during the COVID-19 pandemic. Sampling was based on the results of Norwegian surveys designed for collecting general population norms for health measurement instruments. In December 2019, 12,790 randomly selected adults aged $\geq$18 years received a postal questionnaire. The 3,200 respondents received a similar follow-up postal questionnaire including the EQ-5D-5L, PROMIS-29 instruments, and questions about respondents having or having had COVID-19. Score changes were compared to estimates for the minimal important change (MIC) and age-related change. Association of instrument change scores with baseline characteristics, health problems, and having had COVID-19 was determined using multivariable linear regression. Of 3101 respondents with unchanged addresses, 2423 (78.1%) responded to the second survey. For all respondents, EQ VAS and PROMIS-29 scores for 6 of 8 domains were slightly poorer (p<0.01) than before COVID-19, and the mean change was below the MIC. In multivariable analyses, the greatest number of poorer outcomes were associated with being female, 18–29 years, or $\geq$80 years of age (p<0.01); > MIC for $\geq$ 80 years of age and EQ-5D index, PROMIS-29 physical function and social participation. Respondents who had COVID-19 had poorer outcomes for PROMIS-29 social participation (> MIC). Those reporting COVID-19 in their partner/family and not themselves, had poorer outcomes for PROMIS-29 anxiety and social participation. About 9 months into the COVID-19 pandemic, EQ-5D-5L and PROMIS-29 domain scores showed slightly poorer health in the Norwegian adult general population compared to 1 year earlier in the same respondents. The overall changes were less than expected for age-related change. Relatively poor outcomes defined as important, included general health and social participation for the elderly, and the latter for those having had COVID-19. In conclusion, this study found no evidence for a decline in important aspects of

**Funding:** The study was funded by the Norwegian Research Council (Project Number 262673). Andrew Garratt sought funding and was principal investigator.

**Competing interests:** The authors have declared that no competing interests exist.

adult general population health in Norway that might be attributed to the pandemic at approximately 9 months.

## Introduction

Research into the impact of the COVID-19 pandemic has largely focused on confirmed cases and deaths, rather than general population health and quality of life. General population studies into the effects of the pandemic and public health measures on mental health have reported mixed results across nations and continents [1, 2]. Moreover, the urgent need for such studies may have led to the neglect of appropriate scientific methods with self-selection to online questionnaires contributing to selection bias and other potential methodological problems [3].

Three recent systematic reviews assessed the impact of the pandemic on self-reported mental health and health more generally [1, 2, 4]. The first considered mental health in European general populations and only included longitudinal studies with pre-/early and during pandemic data [1]. The meta-analysis included 27 studies covering 8 countries and found no evidence for changes in anxiety and depression, but emotional distress decreased during the pandemic [1]. The second considered mental health assessed by the Depression, Anxiety and Stress Scale (DASS) in healthy populations and included cross-sectional and longitudinal studies with pre- and during pandemic data [2]. The meta-analysis included 59 studies, 47 countries and comparisons with DASS general population norm data. Depression, anxiety, and stress increased during the pandemic, the former being most prominent. However, differences across continents were found including increases in depression only in European respondents, and no before and during pandemic differences for the US. Overall, females were more adversely affected than males and students more so than the general population [2]. The third review included 25 studies that used seven different patient-reported outcome measures (PROMs) to assess broader aspects of health [4]. It included a meta-analysis of 17 studies for the two most widely reported PROMs, the EQ-5D and WHOQoL-BREF, and concluded that the pandemic had reduced the health and quality of life of the general population across the world.

Additional national surveys include longitudinal studies with pre-pandemic data, and cross-sectional studies during the pandemic with either retrospective measures or norm data for comparative purposes.

For Denmark, pre-pandemic and 1-year data for the Warwick-Edinburgh Mental Well-Being Scale showed poorer mental health, particularly in those with higher education and absence of long-standing health problems [5]. For Japan, a longitudinal survey undertaken in 2020 and 2021 showed poorer SF-8 scores, with the poorest for females and those with a lower education level [6]. For Korea, pre-pandemic data from 2017–2019 and pandemic data for 2020 showed no change in Patient Health Questionnaire-9 depressive symptoms [7]. The EQ-5D-5L was used retrospectively across 13 countries at the end of 2020 and showed poorer scores than before for one third of respondents [8]. The anxiety/depression dimension was most affected, along with females and those under 35 years of age. For Belgium, mental health measurements in a cross-sectional study from May to June 2020 showed poorer emotional health during lockdown than "in general" [9]. For Switzerland, during the first wave of the pandemic in 2020, SF-36 physical and mental health scores were better and poorer than population norms, respectively [10]. For Spain, EORTC QLQ-C30 data collected in April 2020 showed poorer scores for role function, emotional function, and symptom burden compared to population norms [11].

The findings of systematic reviews and other national surveys show contrasting effects of the pandemic on general population health. Those meeting review inclusion criteria largely focused on mental health and are mostly cross-sectional with some recourse to norm data [1, 2, 4]. Further studies that include pre-pandemic data with PROMs that assess broader health effects including aspects of physical and social functioning in representative samples, are important for assessing the full impact of the pandemic on general population health.

Norway had two main epidemic waves of COVID-19 in 2020. Previous studies have defined the first wave from until 17 July 2020 and the second from 18 July 2020 [12, 13]. The second wave ended in February 2021, following nursing homes being opened for more visitors after most residents had been vaccinated with a first dose [14]. The current study includes the longitudinal collection of two PROMs assessing several health domains and general health in a representative sample of the Norwegian general population shortly before and 9 months into the COVID-19 pandemic, the latter coinciding with the second wave.

## Material and methods

### Study design and data collection

This was a longitudinal cohort study of a representative sample of the Norwegian general population who contributed EQ-5D-5L norm data at baseline [15]. The sample size was based on a review of published Norwegian postal surveys collecting similar data, with consideration given to age and sex-specific response rates [15]. The National Registry of the Norwegian Tax Administration (Folkeregisteret) was used to randomly select 12,790 adults aged 18 years and over, who on the 15th of December 2019, were sent an eight-page postal questionnaire and reply-paid envelope addressed to the Norwegian Institute of Public Health (NIPH) [15, 16]. Respondents were sent a second questionnaire and reply-paid envelope on the 11th of November 2020, during the second wave of the COVID-19 pandemic in Norway. The accompanying letters explained the purpose of the study. Both questionnaires included a lottery incentive of ten prizes each to the value of NOK 10,000, approximately 1,000 Euros.

The Regional Committee for Medical and Research Ethics stated that the study did not require their approval. The Data Protection Impact Assessment was approved by the Norwegian Institute of Public Health on 16 October 2019 and 31 August 2020. Written informed consent was received from all respondents to the questionnaire.

### Questionnaire content

The eight-page baseline questionnaire included background questions on demographics (age, sex, education level, civil status, and whether the respondent was born in Norway or not), medical problems through the Self-Administered Comorbidity Questionnaire (SCQ) and two PROMs: the EQ-5D-5L and PROMIS-29 [15, 16]. The SCQ lists 13 medical conditions and up to 3 other non-specified medical problems [16, 17]. The follow-up questionnaire included the EQ-5D-5L, PROMIS-29, but not the SCQ. It also included questions about the respondent, their family, friends, and colleagues having or having had COVID-19 (henceforth referred to as had COVID-19).

The EQ-5D-5L includes five dimensions (mobility, self-care, usual activities, pain/discomfort, anxiety/depression) with five response levels. Health states are transformed to a single index using a scoring algorithm derived from valuation tasks undertaken with general population samples. In the absence of a Norwegian algorithm, Norwegian Medical Products Agency recommendations were followed, including use of the UK value set and mapping [15, 18]. Scores for the EQ-5D index range from -0.59 to 1; 1 is the best possible health state, and negative values represent states worse than dead, which is equal to 0. In addition, the EQ VAS,

assesses self-rated health on a vertical visual analogue scale, with 0–100 endpoints labelled "Worst imaginable. . ." and "Best imaginable health state" respectively. Minimal important differences (MIC) largely in the range 0.024 to 0.100 [19–22] and 5.35 to 10.00 [19–21] have been estimated for the EQ-5D index and EQ VAS, respectively.

The PROMIS-29 is a generic health profile comprising 29 items from the PROMIS domains of anxiety, depression, fatigue, pain (intensity and interference), physical function, sleep disturbance, satisfaction with participation in social roles (social participation) [16, 23]. Each domain comprises four items with five levels except for pain intensity, which has a 0–10 numerical rating scale. The sum of the item responses for each domain are converted to T-scores, where a score of 50 is the average for the US general population with a SD of 10. Higher scores represent more of a domain. Scoring including handling of missing data, was undertaken using the Health Measures Scoring Service application for PROMIS (www.assessmentcenter.net/ac_scoringservice). For PROMIS measures, a MIC of 2 to 6 T-score points was recommended following a systematic review [24]. For the domain of pain intensity, MICs of 2 have been estimated for similar pain numerical rating scales [25, 26]. Lower MIC estimates were used in reporting instrument scores. Respondents were approximately one year older since the first questionnaire. Therefore, change scores below the MIC were compared to age-related change defined as the mean difference in PROMs scores across all ages 18–80 years for norm data [15, 16].

## Statistical analysis

Sample characteristics are presented using mean (SD) or number (%). Descriptive statistics for the EQ-5D and PROMIS-29 are presented using the mean (SD). The distribution of EQ-5D-5L dimension scores before and during the pandemic was compared using the Wilcoxon signed-rank test. Changes on the EQ-5D index, EQ VAS and PROMIS-29 domains and changes were compared overall and in strata according to sex, age categories, education level (collapsed to 2 levels of school or higher education), civil status, born in Norway or not, SCQ medical problems, and having or having had COVID-19. Change scores within respondents were compared using one-sample t-tests.

EQ-5D index, EQ VAS and PROMIS-29 change scores were included as dependent variables in multiple linear regression analysis to assess the contribution of the respondent characteristics listed above. Dependent variable baseline scores were included as an independent variable along with the EQ-5D-5L index scores as a measure of general health. Results of the multiple linear regression analyses are shown with unstandardized beta coefficients and 99% confidence intervals.

Stata (versions 15.0 and 18.0, Stata Corporation, College Station, TX, USA) was used for the statistical analysis. Because of the large number of tests, a 1% level of statistical significance was used with two-sided tests.

## Results

### Response rates and sample characteristics

From 12,363 correctly addressed questionnaires, 3,200 (25.9%) returned a questionnaire that was at least partly completed [15]. The mean age (SD) was 50.9 (20.7), and age ranged from 18 to 97 years (Table 1). Of the 3,101 respondents residing at the same address, 2,423 (77.3%) returned the follow-up questionnaire. Compared to baseline, there was a greater proportion of respondents ≥ 60 years of age at follow-up.

**Table 1. Respondent characteristics[a] at baseline (n = 3,200) and follow-up (n = 2,423).**

|  | Baseline | | Follow-up | |
|---|---|---|---|---|
|  | n | % | n | % |
| Female | 1755 | 55 | 1318 | 55 |
| Male | 1434 | 45 | 1097 | 45 |
| Age, years |  |  |  |  |
| 18–29 | 698 | 22 | 429 | 18 |
| 30–59 | 1226 | 38 | 900 | 37 |
| 60–79 | 999 | 31 | 875 | 36 |
| >=80 | 247 | 7 | 200 | 8 |
| Education |  |  |  |  |
| School | 1536 | 48 | 1116 | 46 |
| Higher education | 1648 | 52 | 1298 | 54 |
| Cohabiting |  |  |  |  |
| No | 1050 | 33 | 750 | 31 |
| Yes | 2130 | 67 | 1661 | 69 |
| Born in Norway |  |  |  |  |
| No | 272 | 9 | 177 | 7 |
| Yes | 2899 | 91 | 2225 | 93 |
| Health problems[b] |  |  |  |  |
| None | 824 | 27 | 602 | 26 |
| One | 809 | 26 | 621 | 27 |
| Two | 668 | 22 | 522 | 22 |
| Three or more | 767 | 25 | 587 | 25 |

[a]Missing data (follow-up): 11 (8), 30 (19), 16 (9), 20 (12), 166 (91) cases for gender, age, education, cohabiting, born in Norway, and health problems respectively.

[b]Self-administered Comorbidity Questionnaire: 13 medical problems and 3 open categories.

## Health outcomes

Fig 1 shows EQ-5D-5L dimension frequency distributions at baseline and follow-up. Pain/discomfort and anxiety/depression showed slightly better and worse outcomes (p<0.01) respectively. The statistically significant differences for EQ VAS and six PROMIS-29 domains showed slightly poorer health at follow-up than at baseline (Table 2). However, all mean changes were less than those found for EQ-5D-5L and PROMIS-29 change scores for a 1-year increase in age and lower than estimates for MIC of the respective scales.

Table 2 also shows follow-up scores for those reporting themselves, partner/family members (not themselves) or colleagues/friends (not themselves or partner/family) as having had COVID-19. Compared to those not reporting COVID-19 across these groups, those who had COVID-19 themselves had significantly poorer outcomes for PROMIS-29 social participation, and the difference was above the MIC. Those whose family members had COVID-19, had significantly poorer outcomes for PROMIS-29 anxiety; below the MIC but above age-related change (1.28 vs. 1.19). For the remainder of the results of the univariate analysis including sex, age groups, education level, SCQ comorbidity, see S1 Table.

The results of the multivariable linear regression analysis show that compared to females, males had significantly better outcomes for the EQ-5D index, and all but PROMIS-29 physical function and sleep disturbance (Tables 3 and 4). Compared to the reference category of 30–59 years of age, the youngest group had poorer outcomes for PROMIS-29 anxiety (> age-related

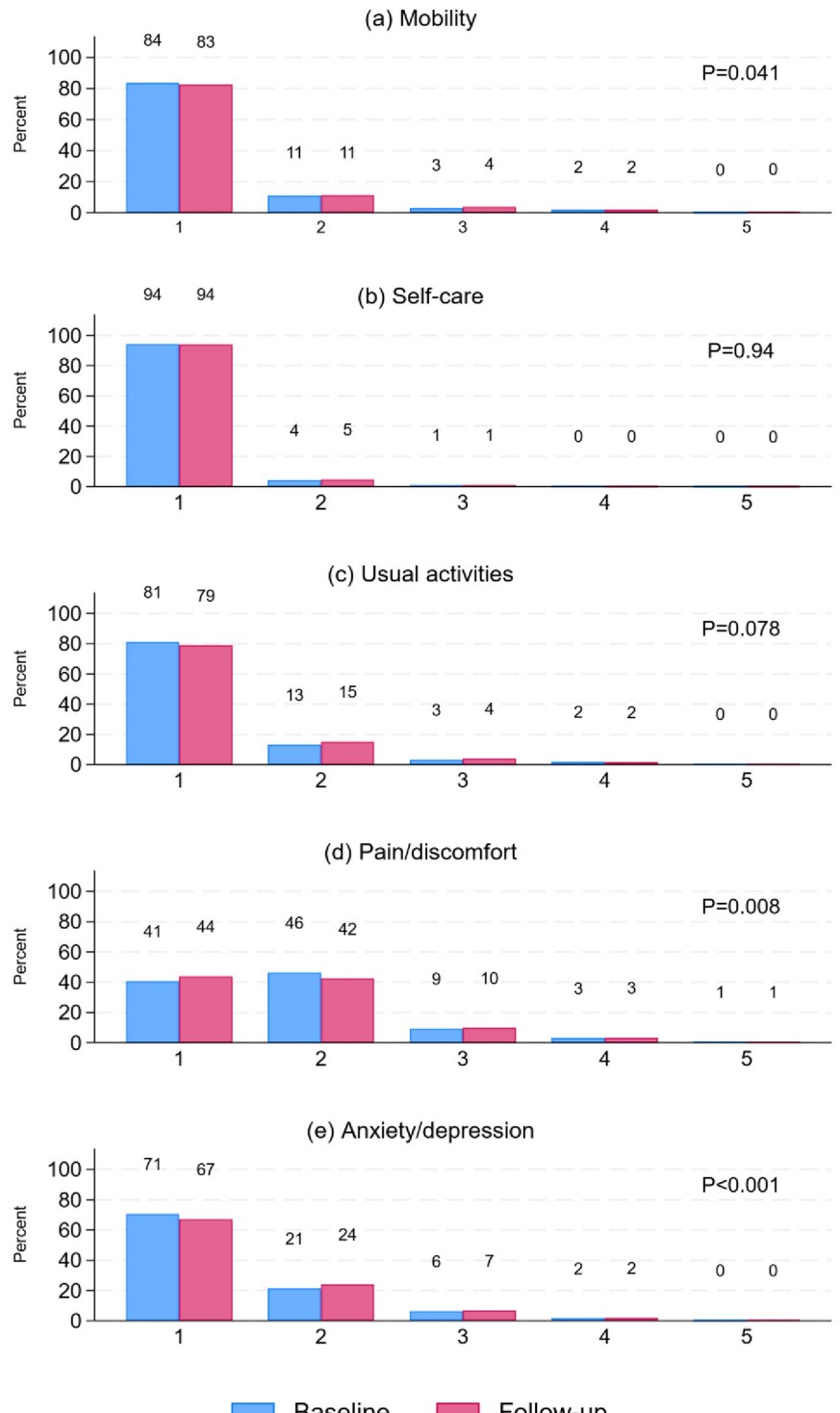

**Fig 1. EQ-5D-5L dimension[a] frequencies at baseline and follow-up.** [a] 1 no problems, 2 slight problems, 3 moderate problems, 4 severe problems, 5 extreme problems / unable to do. P-values: Wilcoxon signed-rank test.

**Table 2. Mean (SD) EQ-5D and PROMIS-29 scores at baseline and follow-up for COVID-19 groups.**

| | All respondents (n = 2,417) | | Follow-up scores for those reporting COVID-19 | | |
| | Baseline | Follow-up | Self (n = 56) | Family/partner (n = 210) | Colleagues/friends (n = 774) |
|---|---|---|---|---|---|
| *EQ-5D-5L index*[a] | 0.832 (0.172) | 0.831 (0.175) | 0.848 (0.195) | 0.826 (0.193) | 0.847 (0.149) |
| *EQ VAS*[b] | 80.49 (16.30) | 79.06 (16.75)* | 75.87 (21.62) | 78.01 (17.95) | 80.82 (14.94) |
| *PROMIS-29*[c] | | | | | |
| Anxiety | 47.38 (8.04) | 47.87 (8.29)* | 46.54 (7.49) | 48.95 (8.53)* | 48.49 (8.34) |
| Depression | 46.89 (7.63) | 47.37 (7.95)* | 47.92 (8.37) | 48.10 (8.43) | 47.37 (7.83) |
| Fatigue | 44.07 (9.51) | 44.07 (9.58) | 43.56 (10.21) | 45.09 (10.20) | 44.18 (9.23) |
| Pain intensity[d] | 1.90 (2.06) | 2.04 (2.06)* | 1.70 (2.05) | 2.05 (2.17) | 1.73 (1.88) |
| Pain interference | 48.60 (8.22) | 49.01 (8.28)* | 48.91 (8.54) | 49.07 (8.40) | 47.86 (7.46) |
| Physical Function | 52.79 (7.30) | 52.53 (7.60)* | 52.40 (8.45) | 52.60 (7.60) | 54.12 (6.12) |
| Sleep disturbance | 47.21 (8.13) | 47.30 (8.09) | 46.53 (7.85) | 47.30 (8.24) | 47.22 (7.99) |
| Social participation | 56.31 (8.46) | 55.30 (9.02)* | 52.40 (10.92)* | 53.99 (9.73) | 55.94 (8.18) |

[a] EQ-5D-5L index scores range from -0.57 to 1 where 1 is the best possible health state.

[b] EQ VAS scores range from 0–100 where 100 is the best possible health state.

[c] T-scores where a score of 50 is the average for the US general population with a standard deviation of 10. Higher scores for domains and items represent more of a domain, for example, higher levels of physical functioning or anxiety.

[d] Numerical rating scale from 0–10; 0 is lowest and 10 the greatest pain intensity.

Asterisks denote statistically significant score changes (column 3) and differences in change scores compared to those not reporting COVID-19 at all compared to those who had COVID-19 themselves (column 4), family/partner had COVID-19 excluding self (column 5), and colleagues/friends had COVID-19 excluding self/partner/family (column 6):

* $p < 0.01$,

** $p < 0.001$.

change for 1 year), depression, and sleep disturbance. Those aged 60–79 years had better outcomes for PROMIS-29 fatigue and poorer outcomes for PROMIS-29 physical function. Those ≥ 80 years of age had poorer outcomes for EQ-5D index (> MIC), EQ VAS (> age-related change), PROMIS-29 anxiety (> age-related change), physical function (> MIC), and social participation (> MIC). Compared to those with school level education, those with a higher education level had better outcomes for PROMIS-29 pain interference. Compared to those not cohabiting, cohabitants had poorer outcomes for PROMIS-29 pain interference. Compared to those not reporting SCQ health problems, those reporting ≥ 3 had poorer outcomes for EQ-5D index (> MIC), EQ VAS (> age-related change), PROMIS-29 pain intensity (> age-related change), pain interference (> MIC), and physical function. Those reporting two health problems had poorer outcomes for PROMIS-29 pain interference. Compared to those not reporting COVID-19 (themselves, partner/family, colleagues/friends), respondents that had it, had poorer outcomes for social participation (> MIC). Those reporting that their partner/family had COVID-19 (not themselves) had poorer outcomes for PROMIS-29 anxiety (> age-related change), and social participation.

## Discussion

Compared to shortly before the COVID-19 pandemic, this study showed little change in specific health domains and general health scores 9 months into the pandemic in the same respondents from a representative sample of the Norwegian general population. Significantly poorer outcomes were found particularly for females, the youngest, and oldest age groups. However, except for physical function and social participation for the latter, these changes

**Table 3. Multiple linear regression of changes in PROMs scores on socioeconomic, baseline health[a] comorbidity and COVID-19 variables (n = 2,256).**

| Independent variables (reference category) | PROMIS-29 domains[d] | | | | | | | | | |
|---|---|---|---|---|---|---|---|---|---|---|
| | EQ-5D index[b] | | EQ VAS[c] | | Anxiety | | Depression | | Fatigue | |
| | Coefficient | 99% CI | Coefficient | 99% CI | Coefficient | 99% CI | Coefficient | 99% CI | Coefficient | 99% CI |
| Male (female) | 0.01* | [0.00, 0.03] | 0.03 | [-1.27, 1.33] | -1.16** | [-1.84, -0.49] | -0.71* | [-1.35, -0.07] | -1.07** | [-1.85, -0.35] |
| Age years (30–59) | | | | | | | | | | |
| 18–29 | -0.01 | [-0.03, 0.01] | -0.37 | [-2.33, 1.60] | 1.51** | [0.50, 2.53] | 1.08* | [0.11, 2.04] | 0.80 | [-0.33, 1.95] |
| 60–79 | -0.00 | [-0.02, 0.01] | 0.18 | [-1.39, 1.76] | -0.48 | [-1.29, 0.33] | -0.16 | [-0.94, 0.62] | -1.69** | [-2.61, -0.75] |
| 80+ | -0.04** | [-0.06, 0.01] | -2.66* | [-5.30, -0.03] | 1.62* | [0.28, 2.96] | 1.28 | [-0.00, 2.56] | 0.04 | [-1.48, 1.53] |
| Higher education (school level) | 0.00 | [-0.01, 0.02] | 0.50 | [-0.84, 1.83] | -0.21 | [-0.90, 0.47] | 0.16 | [-0.49, 0.82] | -0.08 | [-0.86, 0.68] |
| Cohabiting (no) | -0.00 | [-0.02, 0.01] | -0.75 | [-2.26, 0.76] | 0.11 | [-0.66, 0.88] | -0.38 | [-1.12, 0.36] | -0.48 | [-1.36, 0.37] |
| Born outside Norway (born in Norway) | -0.00 | [-0.03, 0.02] | 0.73 | [-1.76, 3.22] | 0.79 | [-0.48, 2.07] | 0.27 | [-0.95, 1.48] | 0.05 | [-1.40, 1.46] |
| SCQ no. health problems (none) | | | | | | | | | | |
| 1 | -0.02 | [-0.03, 0.00] | -0.66 | [-2.46, 1.14] | 0.09 | [-0.83, 1.01] | 0.42 | [-0.46, 1.31] | 0.43 | [-0.61, 1.46] |
| 2 | -0.01 | [-0.03, 0.01] | -0.19 | [-2.16, 1.78] | 0.26 | [-0.74, 1.27] | 0.49 | [-0.47, 1.46] | 0.21 | [-0.94, 1.32] |
| 3+ | -0.04** | [-0.06, -0.02] | -3.23** | [-5.44, -1.03] | 0.29 | [-0.84, 1.41] | 0.71 | [-0.37, 1.78] | 1.24 | [-0.04, 2.51] |
| Had COVID-19 (no) | | | | | | | | | | |
| Respondent | 0.01 | [-0.03, 0.06] | -2.72 | [-7.04, 1.60] | -0.74 | [-2.93, 1.45] | 1.29 | [-0.82, 3.40] | -0.28 | [-2.72, 1.15] |
| Partner/family | -0.01 | [-0.03, 0.01] | -1.27 | [-3.58, 1.03] | 1.42* | [0.24, 2.60] | 0.95 | [-0.18, 2.08] | 1.30 | [0.20, 2.44] |
| Colleague/friend | -0.00 | [-0.021, 0.01] | -0.00 | [-1.44, 1.43] | 0.71 | [-0.03, 1.45] | 0.48 | [-0.23, 1.19] | 0.36 | [-0.51, 1.04] |
| R square | 0.13 | | 0.20 | | 0.17 | | 0.16 | | 0.20 | |

[a] To simplify presentation baseline health scores are not presented. All results were statistically significant (p<0.01). including EQ-5D-5L index (all regressions) plus baseline EQ VAS and PROMIS domain scores corresponding to the dependent variables.

[b] EQ-5D-5L index scores range from -0.57 to 1 where 1 is the best possible health state.

[c] EQ VAS scores range from 0–100 where 100 is the best possible health state. Positive change scores represent improvement.

[d] Domains are T-scores where a score of 50 is the average for the US general population with a standard deviation of 10. Higher scores for domains and items represent more of a domain, for example, higher levels of anxiety or fatigue. Changes scores for anxiety, depression, fatigue,: negative values represent improvement.

Asterisks denote statistically significant differences:

* p<0.01,

** p<0.001.

were below MIC estimates previously reported for those instruments. After controlling for background variables and baseline health, those reporting more than three health problems before the pandemic had poorer outcomes for pain, physical function, and general health. Those who had COVID-19 had poorer scores for social participation which was above the MIC.

An evaluation of Norwegian authorities' handling of the pandemic concluded that it was well undertaken, with little to indicate any sizeable loss of health following the pandemic in Norway [14]. Moreover, Norway performed relatively better across a range of outcomes when compared to Europe and the rest of the world [14]. This is further supported by the overall findings of the current study, including little change in general health 9 months into the pandemic. Much of the population live in remote areas, which may have reduced the rate of transmission and impact of the pandemic. Norway is also a wealthy country, and during this period there was available capacity in hospitals and health care services that might further explain the rather small impact on this general population sample. The elderly and those with health

**Table 4. Multiple linear regression of changes in PROMs scores on socioeconomic, baseline health[a], comorbidity and COVID-19 variables (n = 2,256).**

| Independent variables (reference category) | PROMIS-29 domains[b] | | | | | | | | | |
|---|---|---|---|---|---|---|---|---|---|---|
| | Pain intensity[e] | | Pain interference | | Physical function | | Sleep disturbance | | Social participation | |
| | Coefficient | 99% CI | Coefficient | 99% CI | Coefficient | 99% CI | Coefficient | 99% CI | Coefficient | 99% CI |
| Male (female) | -0.16* | [-0.32, -0.00] | -0.93** | [-1.53, -0.32] | 0.22 | [-0.25, 0.68] | -0.61 | [-1.27, 0.06] | 0.93* | [0.19, 1.67] |
| Age years (30–59) | | | | | | | | | | |
| 18–29 | -0.08 | [-0.32, 0.15] | -0.84 | [-1.75, 0.08] | 0.39 | [-0.32, 1.10] | 1.11* | [0.11, 2.11] | 0.64 | [-0.49, 1.76] |
| 60–79 | 0.10 | [-0.09, 0.28] | 0.19 | [-0.54, 0.92] | -0.60* | [-1.16, -0.04] | -0.06 | [-0.86, 0.74] | -0.11 | [-1.01, 0.79] |
| 80+ | 0.23 | [-0.09, 0.55] | 0.57 | [-0.64, 1.79] | -3.49** | [-4.46, -2.52] | 0.50 | [-0.82, 1.83] | -2.45** | [-3.94, -0.95] |
| Higher education (school level) | -0.14 | [-0.29, 0.02] | -0.66* | [-1.28, -0.03] | -0.01 | [-0.49, 0.47] | 0.16 | [-0.52, 0.85] | -0.09 | [-0.85, 0.67] |
| Cohabiting (no) | 0.17 | [-0.01, 0.35] | 0.73* | [0.03, 1.43] | -0.06 | [-0.60, 0.48] | 0.12 | [-0.65, 0.89] | -0.03 | [-0.89, 0.83] |
| Born outside Norway (born in Norway) | 0.15 | [-0.14, 0.45] | 0.04 | [-1.10, 1.19] | 0.52 | [-0.37, 1.41] | 0.89 | [-0.37, 2.15] | -0.46 | [-1.87, 0.96] |
| SCQ no. health problems (none) | | | | | | | | | | |
| 1 | 0.15 | [-0.06, 0.37] | 0.42 | [-0.42, 1.26] | 0.37 | [-0.28, 1.02] | 0.43 | [-0.49, 1.35] | -0.00 | [-1.03, 1.02] |
| 2 | 0.22 | [-0.02, 0.46] | 1.14* | [0.21, 2.06] | 0.03 | [-0.67, 0.74] | 0.48 | [-0.53, 1.49] | -0.10 | [-1.22, 1.03] |
| 3+ | 0.64** | [0.37, 0.91] | 2.13** | [1.08, 3.19] | -0.83* | [-1.62, -0.04] | 1.11 | [-0.02, 2.23] | -0.88 | [-2.14, 0.38] |
| Had COVID-19 (no) | | | | | | | | | | |
| Respondent | -0.26 | [-0.77, 0.25] | 0.40 | [-1.61, 2.41] | 0.48 | [-1.05, 2.02] | -0.80 | [-2.99, 1.38] | -3.06* | [-5.51, -0.62] |
| Partner/family | 0.02 | [-0.26, 0.29] | 0.11 | [-0.96, 1.18] | 0.03 | [-0.79, 0.86] | 0.08 | [-1.10, 1.25] | -1.38* | [-2.69, 0.06] |
| Colleague/friend | -0.10 | [-0.27, 0.07] | -0.29 | [-0.96, 0.38] | 0.28 | [-0.24, 0.80] | 0.14 | [-0.60, 0.88] | -0.60 | [-1.43, 0.22] |
| R square | 0.18 | | 0.20 | | 0.15 | | 0.21 | | 0.18 | |

[a] To simplify presentation baseline health scores are not presented. All results were statistically significant (p<0.01). including EQ-5D-5L index (all regressions) and PROMIS domain scores corresponding to the dependent variables.

[b] Domains are T-scores where a score of 50 is the average for the US general population with a standard deviation of 10. Higher scores for domains and items represent more of a domain, for example, higher levels of pain interference or physical functioning. Changes scores for pain interference/intensity, sleep disturbance: negative values represent improvement. Change scores for physical function and social participation: positive values represent improvement.

[e] Numerical rating scale from 0–10; 0 is lowest and 10 the greatest pain intensity.

Asterisks denote statistically significant differences:

* p<0.01,

** p<0.001.

problems may have been more vulnerable and affected by administrative measures and social isolation during the pandemic, or may have experienced less attention, which might explain the larger negative impact on these sub-groups. In Norway, vaccination against COVID-19 started on 27 December 2020, therefore, vaccination status is unlikely to influence the results [27].

Two systematic reviews focusing on mental health included 27 European [1] and 59 international [2] studies. The former found no evidence for changes in anxiety and depression, which concurs with findings reported here when compared with MIC estimates and expected age-related change between the two surveys. In the European component of the latter study, depression seemed to increase from pre-/early pandemic to during the pandemic [2]. Females and the young had poorer outcomes under the pandemic, which concurs with findings reported here for PROMIS-29 domains of anxiety and depression. However, the review focused on one instrument, the DASS, and included studies of healthy populations in addition to representative samples of the general population [2]. Another systematic review included PROMs assessing additional

aspects of health, concluding that the pandemic had reduced health and quality of life of the general population internationally [4]. The meta-analysis included the EQ-5D and WHOQOL-BREF, the latter having similar domains as the PROMIS-29 used in the present study. However, only two studies meeting the inclusion criteria included PROMs pre-pandemic, and they showed larger declines in the EQ-5D index of 0.026 and 0.05 for Morocco [28] and Portugal [29], respectively, than those reported here. Additional longitudinal studies with pre-pandemic data also found poorer outcomes for mental health in Danish [5] and general health in Japanese [6] general populations. The latter study also reported larger effect sizes among women than men.

Potential recall bias limits the comparability of an international retrospective study, which found considerably poorer outcomes for the EQ-5D-5L index scores compared to those reported here [8]. This study found that females and the youngest respondents had relatively poorer outcomes concurring with findings reported here. Anxiety/depression and depression was the aspect of health contributing most to poorer outcomes [8]. PROMIS-29 social participation was the only domain that exceeded the MIC in the current study, but PROMIS-29 anxiety showed the next largest effect size for PROMIS-29 scores. Poorer emotional health was also found for general populations based on retrospective data for Belgium in an online survey, where participants were recruited by advertisements on social networks and announcements on university websites [9]. Similar reductions in emotional health have been reported from comparisons with norm data in an online study in a convenience sample in Switzerland [10], and another online study comparing two-cross sectional surveys of participants recruited from online panels with quota sampling in Spain [11]. Because of differences in study design, sample selection, choice of instruments and analytic methods, findings are difficult to compare between studies.

### Strengths and limitations

Compared to the majority of studies assessing the impact of the pandemic on general population health [1, 2, 4], the study has several important strengths. First, the before and during the pandemic longitudinal design allowed the measurement of health outcomes in the same respondents during the pandemic. The Norwegian norm data collected for the EQ-5D-5L, meant that an opportunistic study was possible under the pandemic using the same methods of data collection for the follow-up survey [15]. The original sample was a random sample of the general Norwegian population ≥ 18 years of age and the data should be more representative than studies based on convenience samples or self-selected populations recruited through social networks or advertising.

The EQ-5D-5L and PROMIS-29 are both generic PROMs and include aspects of health that are of broad importance for the general population and across health problems. They give complementary information and MIC estimates are available [24]. In addition, the study used norm data from the original baseline population as a means of accounting for average changes in PROMs scores based on age differences of respondents at 1 year [15]. This was considered necessary given the small changes expected in EQ-5D-5L and PROMIS-29 scores.

Study limitations include the 1-year follow-up period, which only covers the first 9 months of the pandemic in Norway, i.e., the first two waves of the pandemic. It is possible that short-term health outcomes were less favorable during the initial phase of the pandemic and an additional questionnaire 3–6 months after the first, would have been necessary to assess this. The timing of the pandemic survey at the end of 2020 overlapped with those included in systematic reviews, but several included early and late pandemic surveys [1, 2, 4]. Moreover, we do not have dates of when the respondents had COVID-19, and linkage with national registers of PCR-positive was not possible.

The low response rate of 26% to the original survey which collected norm data for the EQ-5D-5L, was expected following findings from similar Norwegian surveys [15]. Studies have found decreasing response rates over time, but other important factors include the survey population, research topic, questionnaire length, and use of incentives [30]. Both the baseline and follow-up used a traditional postal survey which tend to have higher response rates compared to web-surveys [31]. Postal reminders were not used and may have increased response rates.

The two brief generic PROMs might not have fully captured the full impact of the pandemic on quality of life, and some studies included COVID-19 pandemic-specific PROMs [4, 29] and mental health instruments that might be relevant to the pandemic [1, 2]. The inclusion of the SCQ and several background questions meant that outcomes for potentially vulnerable groups could be assessed, but sample sizes were small, particularly in relation to several SCQ medical problems.

## Conclusions

This study found that 9 months into the COVID-19 pandemic, the health of the Norwegian adult general population had slightly deteriorated as assessed by the EQ-5D-5L and PROMIS-29 domain scores. Poorer health outcomes were associated with being female, the youngest and oldest age groups, several medical problems, and having had COVID-19, or a partner/family member who had COVID-19. According to MIC estimates, those experiencing the poorest health outcomes 9 months into the pandemic were over 80 years of age, had several health problems pre-pandemic, and had COVID-19. Cost-effective interventions for potentially vulnerable groups including the elderly and those with multi-morbidity are needed.

## Supporting information

**S1 Table. Mean (SD) change in EQ-5D and PROMIS-29 scores for socioeconomic groups and self-reported health problems.**
(DOCX)

## Acknowledgments

Kirsten Danielsen, Inger Paulsrud and Kjetil Telle contributed to organizational aspects of the survey. Inger Paulsrud was responsible for scanning the questionnaires and along with Olaf Holmboe, preparing the data file.

## Author Contributions

**Conceptualization:** Andrew M. Garratt, Knut Stavem.

**Data curation:** Andrew M. Garratt.

**Formal analysis:** Andrew M. Garratt, Knut Stavem.

**Funding acquisition:** Andrew M. Garratt.

**Investigation:** Andrew M. Garratt.

**Methodology:** Andrew M. Garratt, Knut Stavem.

**Project administration:** Andrew M. Garratt.

**Writing – original draft:** Andrew M. Garratt.

**Writing – review & editing:** Andrew M. Garratt, Knut Stavem.

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
