## [Decision Letter · Decision Letter 0]

23 Aug 2024

PONE-D-24-18493COVID-19 and self-reported health of the Norwegian adult general population: a longitudinal study 3 months before and 9 months into the pandemicPLOS ONE

Dear Dr. Garratt,

Thank you for submitting your manuscript to PLOS ONE. After careful consideration, we feel that it has merit but does not fully meet PLOS ONE’s publication criteria as it currently stands. Therefore, we invite you to submit a revised version of the manuscript that addresses the points raised during the review process.

We look forward to receiving your revised manuscript.

Kind regards,

Esteban Ortiz-Prado

Academic Editor

PLOS ONE

“The study was funded by the Norwegian Research Council (Project Number 262673).  Andrew Garratt sought funding and was principal investigator.”

Additional Editor Comments:

I hope you are doing well. I am writing to inform you that your manuscript titled "COVID-19 and self-reported health of the Norwegian adult general population: a longitudinal study 3 months before and 9 months into the pandemic" (PONE-D-24-18493) has been reviewed and is now accepted for publication in PLOS ONE, pending minor revisions.

Two of the reviewers have provided positive feedback, recommending the acceptance of your manuscript in its current form. They appreciated the comprehensive nature of your study and the rigorous methods employed.

Reviewer 1 has suggested a few minor revisions to enhance the clarity and contextual depth of your manuscript. These suggestions are intended to further strengthen the presentation of your findings.

Please review the comments provided by Reviewer 1 and make the necessary revisions to your manuscript. Once you have addressed these points, kindly resubmit your revised manuscript along with a detailed response letter outlining how each comment has been addressed.

We are excited to see your work published and look forward to receiving the final version of your manuscript.

Thank you for your valuable contribution to the journal.

Best regards,

Esteban Ortiz-Prado, PhD

Guest Editor

PLOS ONE

Reviewers' comments:

Reviewer's Responses to Questions

**Comments to the Author**

1. Is the manuscript technically sound, and do the data support the conclusions?

Reviewer #1: Yes

Reviewer #2: Yes

Reviewer #3: Yes

2. Has the statistical analysis been performed appropriately and rigorously? 

Reviewer #1: Yes

Reviewer #2: Yes

Reviewer #3: Yes

3. Have the authors made all data underlying the findings in their manuscript fully available?

Reviewer #1: Yes

Reviewer #2: Yes

Reviewer #3: Yes

4. Is the manuscript presented in an intelligible fashion and written in standard English?

Reviewer #1: Yes

Reviewer #2: Yes

Reviewer #3: Yes

5. Review Comments to the Author

Reviewer #1: The title of the study focuses on an important societal issue that deserves further scholarly attention. As noted in the separate review template, this essay would benefit from adding more context and evidence to fully support its core argument.

Reviewer #2: I liked the attempt by the authors to obtain a sample which is representative of the Norwegian general adult population to track the self-reported health and Covid-19 outcomes. I also appreciated the author's use of validated data collection and analysis methods in this study.

Reviewer #3: The study presents a comprehensive summary of the health impacts of the COVID-19 pandemic on the Norwegian general population. It is well-structured and provides key details on the methodology, findings, and conclusions.

6. PLOS authors have the option to publish the peer review history of their article (what does this mean?). If published, this will include your full peer review and any attached files.

Reviewer #1: No

Reviewer #2: **Yes: **Kenneth Kudzai Maeka

Reviewer #3: **Yes: **Kawalpreet Kaur

---

## [Author Response · Author response to Decision Letter 0]

20 Sep 2024

Review report on paper titled “COVID-19 and self-reported health of the Norwegian adult general population: a longitudinal study 3 months before and 9 months into the pandemic”

Editor’s Comments

Abstract 

1. What is the motivation behind investigating this specific area?

The first sentence of the abstract now explicitly states that the study aim was to assess the impact of the pandemic on broader aspects of health in the general population (Abstract, lines 17-18).

2. Which sampling method was utilized?

We have now stated that sampling for the baseline study followed the results of similar Norwegian surveys designed for the collection of general population norms. We have given the sample size and stated that they were randomly selected (Abstract, lines 20-21). 

3. What are the key conclusions drawn?

We have included a final sentence in the abstract which concludes that based on the study findings, there was no evidence for a decline in important aspects of general population health (Abstract, lines 39-41). 

4. Can you provide some keywords that encapsulate the essence of the study?

Keywords now include COVID-19, EQ-5D-5L, general population, PROMIS-29, survey (line 43). 

Introduction 

5. The duration of the Covid-19 pandemic in Norway and its impact on the Norwegian adult population are important topics to explore. To effectively engage readers, it is vital to present compelling evidence that showcases the duration of the pandemic in the area. Does it really spanned over duration of 9-months?

We have now included information on the duration of the first and second waves of the pandemic in Norway which are most relevant to the article. We have also included information on the government response including lockdown measures, together with references (page 5, lines 93-96; additional references 12-14). We have also stated that the follow-up survey coincided with the second wave (page 6, lines 98-99).

6. Rephrase “In November 2020, 3200 respondents to a representative general population survey in Norway in December 2019 received a follow-up postal questionnaire including the EQ-5D-5L, PROMIS-29 instruments, and questions about respondents having or having had COVID-19”

This has been rephrased following further information being given about sampling in response to the point above (Abstract, lines 21-24). 

Materials and methods

7. Statistical Power for sample size? 

We have now stated the sample size for the published baseline component of the study was based on a review of existing Norwegian studies that had similar aims. The reference used is already included in the reference list (page 6 lines 104-106). 

8. Is there a difference in the content of the questionnaires used in baseline and follow-up studies? Clarity is needed regarding whether the questionnaire content should remain consistent or be varied.

We have explicitly stated that the EQ-5D-5L, PROMIS-29 and SCQ were included at baseline. We have also stated that the SCQ was not included and that the COVID-19 related questions were included at follow-up (pages 6-7, lines 123-127). 

9. What particular strategies did the authors utilize to control the influence of variables that could impact the minimal important change (MIC) and age-related changes? Or how could you specifically relate health outcomes with COVID-19 only?

Given that this was a general population cohort study, it was only possible to control for the variables collected by means of the two questionnaires including baseline health and respondent characteristics, including age, as independent variables in the multivariate regression analyses (page 8, lines 165-167; Tables 3-4). 

The minimal important change (MIC) is the is the smallest change above which individuals themselves perceive as important (reference 24). This follows from the measurement properties of the instruments as well as the study population, although there is no universal method for deriving MICs. This study was not designed to determine the MIC, as we had no external anchors to determine whether changes were meaningful or not. We therefore used available MIC from other studies, as reported in the Methods (references 19-24). In the Discussion, we have added that we compare with previously reported MIC estimates for these instruments (page 14, lines 224-225).

10. Were there any specific methods employed by the authors to follow the participants throughout the study? Or to overcome the challenges due to participants dropping out over time, which can affect the validity of the results.

Reminders were not used at baseline or follow-up but may have helped increase the response rates slightly. We used a lottery incentive at baseline and continued with this at follow-up. We have extended the strengths and limitations section of the discussion accordingly (page 17, lines 300-301). 

11. How could authors overcome or capture short-term health outcomes?

We might have assessed short-term health outcomes by means of an additional questionnaire during the first wave of the pandemic. We have now stated this in the discussion (page 16, lines 291-292). 

Discussion 

12. The discussion fails to provide an explanation for the potential factors influencing the results; rather, it simply compares them to previous findings, emphasizing the need for a revision.

We have added a new paragraph to the discussion which gives and explanation of factors which may have influenced the results. The findings are in line with a report commissioned by the Norwegian government. We have included a reference [14] along with consideration to when vaccination against COVID-19 started in Norway (page 14, lines 229-240) with an additional reference [27]. 

Strengths and limitations 

13. Too long 

This section was shortened without loss of specific study strengths and limitations. We have shortened the strengths component relating to aspects of data collection (page 16, lines 274-288). We have shortened the limitations component relating to generic and specific patient-reported outcome measures (page 17, lines 302-304). 

Recommendation

14. Needed.

At the close of the conclusions, we have stated that interventions are needed for potentially vulnerable groups (page 17, lines 315-316).

---

## [Editor Report · Decision Letter 1]

3 Oct 2024

COVID-19 and self-reported health of the Norwegian adult general population: a longitudinal study 3 months before and 9 months into the pandemic

PONE-D-24-18493R1

Dear Dr. Garratt,

We’re pleased to inform you that your manuscript has been judged scientifically suitable for publication and will be formally accepted for publication once it meets all outstanding technical requirements.

Kind regards,

Esteban Ortiz-Prado

Academic Editor

PLOS ONE

Additional Editor Comments (optional):

Dear Authors,

Thank you for your revised submission. After reviewing the manuscript, I believe the study is well-constructed and adds valuable insights into the health of the Norwegian general population during the COVID-19 pandemic. The reviewers' comments have been adequately addressed, and the manuscript has significantly improved.

However, there are still minor revisions that need to be addressed before final acceptance. I encourage the authors to make sure that all suggested changes by the reviewers are carefully incorporated into the manuscript. Once these are completed, I am confident the manuscript will be ready for acceptance.

Thank you for your efforts, and I look forward to receiving the final version.

Best regards,

Esteban Ortiz Prado
---

## [Editor Report · Acceptance letter]

15 Oct 2024

PONE-D-24-18493R1 

PLOS ONE

Dear Dr. Garratt, 

I'm pleased to inform you that your manuscript has been deemed suitable for publication in PLOS ONE. Congratulations! Your manuscript is now being handed over to our production team.

Kind regards, 

on behalf of

Dr. Esteban Ortiz-Prado 

Academic Editor

PLOS ONE